# Place-provider-matrix of bystander cardiopulmonary resuscitation and outcomes of out-of-hospital cardiac arrest: A nationwide observational cross-sectional analysis

Dae Kon Kim[1], Sang Do Shin[2]*, Young Sun Ro[3], Kyoung Jun Song[4,5], Ki Jeong Hong[4], So Yeon Joyce Kong[3]

1 Department of Emergency Medicine, Seoul National University Bundang Hospital, Seoul, Republic of Korea, 2 Department of Emergency Medicine, Seoul National University College of Medicine, Seoul, Republic of Korea, 3 Laboratory of Emergency Medical Services, Seoul National University College of Medicine, Seoul, Republic of Korea, 4 Department of Emergency Medicine, Seoul National University Boramae Medical Center, Seoul, Republic of Korea, 5 Laboratory of Emergency Medical Services, Seoul National University Hospital Biomedical Research Institute, Seoul, Republic of Korea

* shinsangdo@gmail.com

**Data Availability Statement:** Data cannot be shared publicly because of patient confidentiality. Data are available from the Korea Center for

## Abstract

### Aims

This study aims to test the association between the place-provider-matrix (PPM) of bystander cardiopulmonary resuscitation (CPR) and outcomes of out-of-hospital cardiac arrest (OHCA).

### Methods

Adult patients with OHCA with a cardiac etiology from 2012 to 2017 in Korea were analyzed, excluding patients who had unknown information on place, type of bystander, or outcome. The PPM was categorized into six groups by two types of places (public versus home) and three types of providers (trained responder (TR), family bystander, and layperson bystander). Outcomes were survival to discharge and good cerebral performance category (CPC) of 1 or 2. Multivariable logistic regression analysis was performed to test the association between PPM group and outcomes with adjustment for potential confounders to calculate adjusted odds ratios (AORs) and 95% confidence intervals (CIs) (reference = Public-TR).

### Results

A total of 73,057 patients were analyzed and were categorized into Public-TR (0.6%), Home-TR (0.3%), Public-Family (1.8%), Home-Family (79.8%), Public-Layperson (9.9%), and Home-Layperson (7.6%) groups. Compared with the Public-TR group, the AORs (95% CIs) for survival to discharge were 0.61 (0.35–1.05) in the Home-TR group, 0.85 (0.62–1.17) in the Public-Family group, 0.38 (0.29–0.50) in the Home-Family group, 1.12 (0.85–1.49) in the Public-Layperson group, and 0.42 (0.31–0.57) in the Home-Layperson group.

Disease Control and Prevention. Institutional Data Access / Ethics Committee (contact via timthe@gmail.com) for researchers who meet the criteria for access to confidential data.

**Funding:** This study was supported by the National Emergency Management Agency of Korea and the Korea Centers for Disease Control and Prevention (CDC). The study was funded by the Korea CDC (2012-2016) (Grant No: 2012-E33010-00; 2013-E33015-00; 2014-E33011-00, 2016-E33012-00, and 2017-Private Support Grant).

**Competing interests:** No authors have competing interests

The AORs (95% CIs) for good CPC were 0.58 (0.27–1.25) in the Home-TR group, 0.88 (0.61–1.27) in the Public-Family group, 0.38 (0.28–0.52) in the Home-Family group, 1.20 (0.87–1.65) in the Public-Layperson group, and 0.42 (0.30–0.59) in the Home-Layperson group.

## Conclusion

The OHCA outcomes of the Home-Family and Home-Layperson groups were worse than those of the Public-TR group. This finding suggests that OHCA occurring in private places with family or layperson bystanders requires a new strategy, such as dispatching trained responders to the scene to improve CPR outcomes.

## Introduction

Out-of-hospital cardiac arrest (OHCA) causes severe mortality around the world. Approximately 30,000 South Koreans, 300,000 Europeans, and 400,000 Americans experience OHCA each year, with survival rates of 3.0% in Korea, 10.3% in Europe and 9.5% in the United States. [1–3]

Rapid cardiopulmonary resuscitation (CPR) and public-access defibrillation (PAD) are the most important components of community CPR programs in treating OHCA. The potential capacity of the bystander is critical for providing rapid CPR and defibrillation. [4] There are three main types of bystanders: trained responders (firefighters, security guards, police officers), family members, and non-family laypersons. Trained responders (TRs) refer to a specific group of individuals who have a high chance of encountering cardiac arrest in daily life because of occupational characteristics and training for CPR situations but who are not part of the officially organized emergency response system in a community. In general, TRs fall between a bystander and first responder according to the Utstein definition of provider. TR services have been regarded as the most basic fundamental programs of the modern emergency medical services (EMS) system. [5, 6] However, TR systems, as a component of community CPR programs, are not well established as part of EMS systems in many countries.

Usually, family or layperson bystanders have little CPR experience, greater fear of performing CPR, and more anxiety regarding legal responsibility than TRs. [7, 8] The quality of CPR by a layperson may be different according to an individual's self-efficacy in performing CPR. Thus, the quality of CPR cannot be assured to the same extent that it can be when performed by TRs. [4, 9] The type of bystander (TR, family member, and layperson) in the community CPR program is one of the most critical factors related to outcomes after OHCA. Another important factor related to outcomes of OHCA is the place of the cardiac arrest event. [10, 11] OHCA occurring in public places is typically characterized by younger patient age, more ventricular fibrillation (VF) in the initial ECG rhythm, and more attempted CPR by bystanders. Therefore, the final outcomes are much better in OHCA occurring in public places than in OHCA occurring at home. [11]

The place-provider-matrix (PPM) in the community CPR program is a new conceptual framework for stratifying outcomes of OHCA based on a combination of place and provider factors. This matrix combines two important aspects of OHCA: the event place (row) and the bystander characteristic (column), which constitute the basis of rapid CPR and early defibrillation. This framework can be used to evaluate the performance of community CPR programs and outcomes of OHCA more effectively.

We hypothesized that the Public-TR group would exhibit the best survival and good neurologic outcome and that different PPM groups would show worse outcomes. The goal of this study is to compare the demographic findings according to the place-provider matrix and to test whether the PPM group is associated with outcomes after OHCA. In addition, we compared the effect size of the PPM group according to the time of OHCA incidence between day and night.

## Methods

### Study design and setting

This is a retrospective, nationwide, multicenter and cross-sectional study using the national OHCA registry from 2012 to 2017.

### Study setting

According to the 2004 National Emergency Medical Services (EMS) Act, school teachers, sports instructors, public transportation vehicle drivers, safety guards of national parks, and policemen are required to receive CPR education to encourage bystander CPR. In addition, private places where TRs work or reside were designated as mandatory sites for PAD programs in 2008 and 2011 by the EMS Act. This group with mandatory CPR education is referred to as trained responders (TRs). TRs have been required to complete a regular two-hour course of CPR training at least once a year since 2005. [6] TRs are not first responders who respond to medical emergencies in an official capacity as part of an organized medical response team. Rather, they are similar to bystanders but have been trained in CPR because of the higher chance of encountering CPR situations due to occupational characteristics. Therefore, TRs do not have the duty of official calls or dispatches from EMS systems. They voluntarily participate in the CPR situation.

The Korean EMS are based on a single-tiered, fire-based, and government-sponsored system. EMS support a population of approximately 50 million people and provide a basic to intermediate level of ambulance services in sixteen provinces. Emergency Medical Technicians (EMT) can provide CPR at the scene and during transport with automatic external defibrillation (AED) and advanced airway management under direct medical control. Advanced cardiac life support (ACLS) drugs are available in the emergency department (ED) and are limited in most prehospital areas of the country. The PAD program started in 2009, but it was not widely used until recently. [12]

The EDs in Korea are classified into three levels according to resources and functional requirements set by the national government. Level 1 (n = 19) and level 2 (n = 110) EDs have more resources and better facilities for emergency care and emergency physicians to manage patients 24 hours a day and 365 days a year. Level 3 EDs (n = 310) can be staffed by general physicians. The CPR guidelines of international academic societies are generally used and are recommended in clinical practice and research. The 2015 AHA guidelines have recently been accepted as the standard guideline by national academic organizations. [13, 14]

### Data source and collection

This study used the Korean OHCA registry constructed from two databases, which included all OHCAs transported by the EMS since 2006. [3, 9, 15] One database consists of an EMS cardiac arrest registry recorded by the EMS providers of the National Fire Agency, and the other consists of a hospital cardiac arrest registry for hospital care and outcomes collected by the Korea Centers for Disease Control and Prevention (CDC). The OHCA case documentation

was sent to the Korea CDC and matched with hospital medical records created by trained medical record reviewers who evaluated all hospital records related to the care provided in the ED, intensive care unit, and wards, as well as the outcomes at discharge. The data quality management team, consisting of EMS physicians, epidemiologists, biostatistics experts, and cardiologists, meets monthly and maintains the data quality through education and feedback to medical record reviewers regarding unclear variables during medical record reviews. [6, 13]

This study was reviewed and approved by the institutional review board of the study hospital. The requirement for informed consent was waived because the data variables did not include personal information, and the study process did not result in any risk for patients. [6] The Korea CDC approved the use of the national registry for this study.

## Study population

All cases of OHCA of presumed cardiac etiology, with a patient age older than 18 years old, and with CPR attempted by EMS providers from January 2012 to December 2017 were included. A cardiac etiology was presumed in the absence of any other obvious cause, such as trauma, drowning, hanging, overdose, or asphyxia, according to clinical information. Patients whose OHCA was witnessed by EMS providers and who collapsed during ambulance transport or in medical facilities or nursing homes were excluded. Patients with missing information regarding the place of the event, bystander characteristics, and outcomes were excluded.

## Variables

The main exposure was the PPM classification, which included the first CPR or defibrillation provider (trained responder, family bystander, and non-family layperson bystander) with stratification by place (public or home). The TR category included firefighters, policemen, public transportation vehicle drivers, school health teachers, sports facility employees, lifeguards, workplace safety employees, travel business employees, and those designated by the EMS Act and regulations. [6] A family bystander (Family) was defined as a family member of the patient. Non-family layperson bystanders (Laypersons) included nearby bystanders, colleagues, friends, and other bystanders. The public places included a public/commercial building, a street/highway, an industrial place, a transport center, a recreation place and a farm. Home places included home residences and dormitories. The PPM was divided into six groups: group 1 (Public-TR), group 2 (Home-TR), group 3 (Public-Family), group 4 (Home-Family), group 5 (Public-Layperson) and group 6 (Home-Layperson).

The variables were general factors, Utstein factors and EMS factors. General factors included gender, patient age, date/time factors such as year, season, weekend, and time. Utstein factors included metropolis, witness, bystander CPR and defibrillation, and primary ECG (shockable versus non-shockable). [16] EMS factors included response time interval (RTI), airway management, EMS defibrillation, and hospital factors (ED level 1 to 4).

## Outcome measure

The primary outcome was favorable neurological outcome, defined as cerebral performance category (CPC) scale score of 1 (good cerebral performance) or 2 (moderate cerebral disability) at hospital discharge. The secondary outcome was survival to discharge. The CPC and survival to discharge were determined according to a review of hospital discharge abstract records of the Korea CDC medical record review.

## Statistical analysis

A descriptive analysis was performed to examine the distribution of potential risk factors for outcomes among each PPM group. The categorical variables were described using counts and proportions and compared with the chi-square test. The continuous variables were compared using the Wilcoxon rank sum test. A multivariable logistic regression analysis was performed to test the association between the PPM group and outcomes (reference = Public-TR group). Potential confounders, such as gender, age, year, season, weekend, metropolis, witness, bystander CPR, bystander AED, primary ECG, RTI, airway, EMS defibrillation, and ED level, were adjusted. The adjusted odds ratios (ORs) and 95% confidence intervals (CIs) were calculated for outcomes. We performed an interaction analysis to compare the effect size of the arrest time, daytime (06:00–17:59) versus nighttime (18:00–05:59), on the outcomes in the final multivariable logistic models. All analyses were performed using SAS version 9.4 (SAS© Cary, NC, USA).

## Results

### Demographic findings

Of the 171,534 eligible OHCA patients, 73,057 patients were finally analyzed, excluding patients with noncardiac etiology (n = 45,926), who did not meet the age criteria (n = 1,903), who were not treated by EMS (N = 11,320), with unknown arrest place (N = 11,752), who experienced arrest during ambulance transport (N = 6,957), who collapse in medical facilities/nursing homes (N = 10,141), with unknown bystander information (N = 7,111), with arrest witnessed by a healthcare provider (N = 3,367) and with unknown outcome (N = 0) (Fig 1).

Of these 73,057 patients, the number (percent) of patients in each PPM group was 438 (0.6%) in the Public-TR group, 188 (0.3%) in the Home-TR group, 1,319 (1.8%) in the Public-Family group, 58,307 (79.8%) in the Home-Family group, 7,226 (9.9%) in the Public-Layperson group, and 5,579 (7.6%) in the Home-Layperson group. The survival to discharge and good CPC rate of each PPM group were 17.8%/11.6% in the Public-TR group, 8.0%/4.8% in the Home-TR group, 15.5%/10.6% in the Public-Family group, 4.9%/2.8% in the Home-Family group, 21.0%/15.1% in the Public-Layperson group, and 6.1%/3.5% in the Home-Layperson group, respectively (Table 1 and Fig 2). Table 2 describes the demographic findings of variables according to arrest time. The survival to discharge and good CPC rates were 7.7% and 4.9% during the nighttime and 6.4% and 3.9% during the daytime, respectively.

### Main analysis

Compared to the Public-TR group, the AORs (95% CIs) by PPM group for survival to discharge were 0.61 (0.35–1.05) in the Home-TR group, 0.85 (0.62–1.17) in the Public-Family group, 0.38 (0.29–0.50) in the Home-Family group, 1.12 (0.85–1.49) in the Public-Layperson group, and 0.42 (0.31–0.57) in the Home-Layperson group. The Home-Family and Home-Layperson groups had significantly worse results than the Public-TR group for survival to discharge. The AORs (95% CIs) by PPM group for good CPC were 0.58 (0.27–1.25) in the Home-TR group, 0.88 (0.61–1.27) in the Public-Family group, 0.38 (0.28–0.52) in the Home-Family group, 1.20 (0.87–1.65) in the Public-Layperson group, and 0.42 (0.30–0.59) in the Home-Layperson group. The Home-Family and Home-Layperson groups also had significantly worse results than the Public-TR group for good CPC (Table 3). The AORs (95% CIs) of survival to discharge rate and good CPC rate in the daytime were 0.85 (0.80–0.91) and 0.84 (0.77–0.91), respectively. The outcome results were found to be better at nighttime than in the daytime after adjusting for confounders (Table 4).

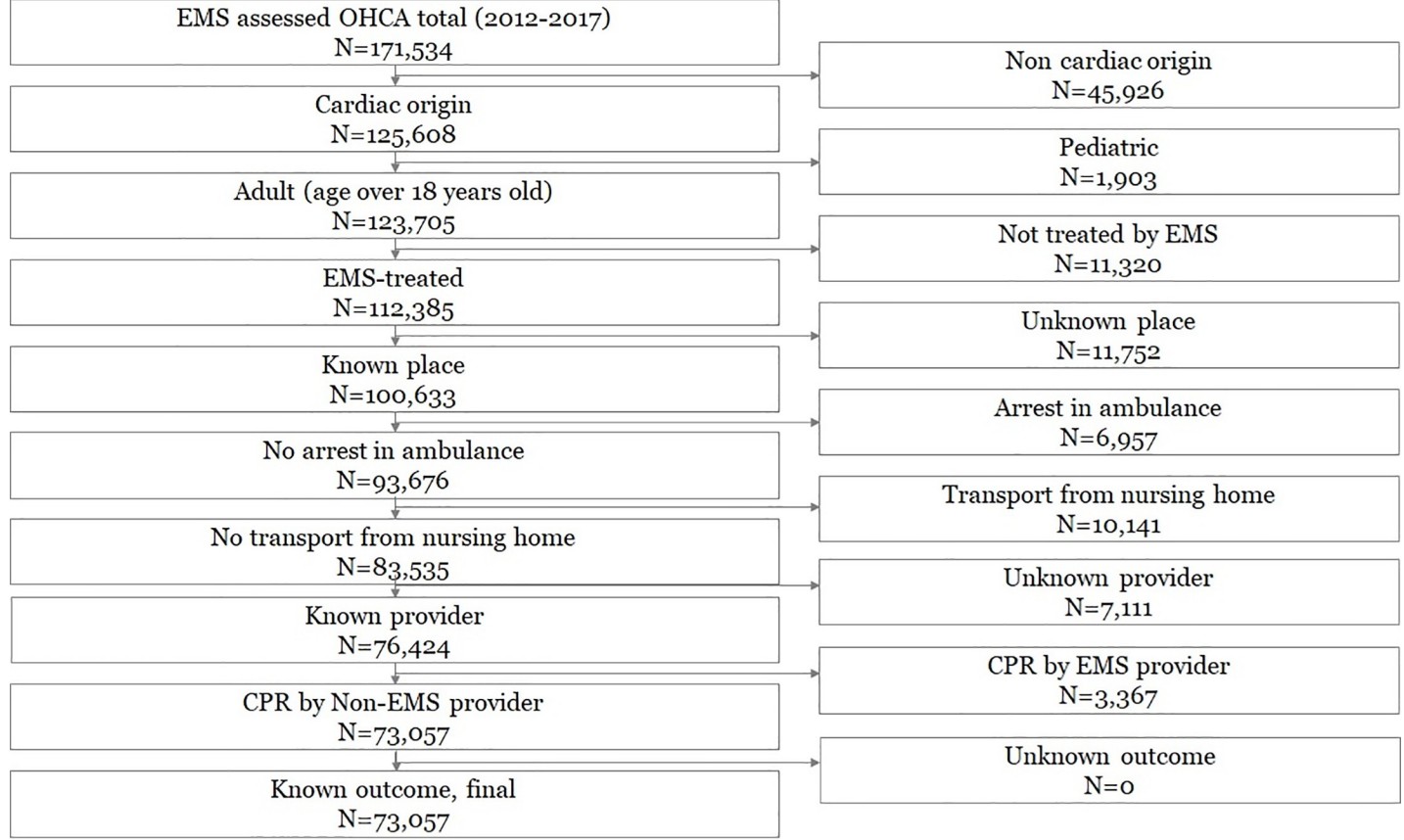

**Fig 1. Study flow chart.** OHCA, out-of-hospital cardiac arrest; EMS, emergency medical services; CPR, cardiopulmonary resuscitation.

### Interaction analysis

In the interaction model, the AORs (95% CIs) in the Public-Family group were 1.10 (0.90–1.35) in the daytime and 1.28 (1.06–1.53) in the nighttime (18:00–06:00). The other PPM groups did not show significantly different differences between the daytime and nighttime (Table 5).

### Discussion

In this study, we discovered that OHCAs that occurred in public places with TRs showed a better OR of survival to discharge and good CPC than those in Home-family and Home-Layperson groups. This result partially corresponded to the study hypothesis. The fact that home places with family and laypersons had lower ORs supported the study hypothesis. However, the fact that the Public-TR, Public-Layperson, Public-Family, and Home-TR groups did not show significant differences was not expected according to the study hypothesis.

It is already known that approximately 80% of OHCAs worldwide occur in private places, such as home residences. [16] However, a higher survival rate and good neurological outcome have been reported in OHCAs occurring in public places. [17] Our study results also correspond to previous findings of other studies. [18] In this study, both outcomes showed higher rates in public places than in home settings, which was independent of bystander type. This fact was shown in our study results showing that there was no significant difference in AORs

**Table 1. Demographics of the study population according to the place-provider-matrix.**

| Variables | | All | | Place-provider-matrix | | | | | | | | | |
| --- | --- | --- | --- | --- | --- | --- | --- | --- | --- | --- | --- | --- | --- |
| | | | | Trained responder | | | | Family bystander | | | | Layperson bystander | | | |
| | | | | Public | | Home | | Public | | Home | | Public | | Home | |
| | | N | % | N | % | N | % | N | % | N | % | N | % | N | % |
| All | | 73057 | 100.0 | 438 | 100.0 | 188 | 100.0 | 1319 | 100.0 | 58307 | 100.0 | 7226 | 100.0 | 5579 | 100.0 |
| Gender | | | | | | | | | | | | | | | |
| | Female | 25719 | 35.2 | 82 | 18.7 | 52 | 27.7 | 414 | 31.4 | 21820 | 37.4 | 1194 | 16.5 | 2157 | 38.7 |
| | Male | 47338 | 64.8 | 356 | 81.3 | 136 | 72.3 | 905 | 68.6 | 36487 | 62.6 | 6032 | 83.5 | 3422 | 61.3 |
| Age | | | | | | | | | | | | | | | |
| | Age<65 | 26923 | 36.9 | 292 | 66.7 | 101 | 53.7 | 665 | 50.4 | 18634 | 32.0 | 4755 | 65.8 | 2476 | 44.4 |
| | 65 = <Age<85 | 36569 | 50.1 | 130 | 29.7 | 74 | 39.4 | 574 | 43.5 | 31092 | 53.3 | 2255 | 31.2 | 2444 | 43.8 |
| | Age> = 85 | 9565 | 13.1 | 16 | 3.7 | 13 | 6.9 | 80 | 6.1 | 8581 | 14.7 | 216 | 3.0 | 659 | 11.8 |
| | Median (q1-q3) | 71 (58–80) | | 59 (50–69) | | 63 (53–76) | | 64 (54–75) | | 73 (60–81) | | 58 (50–70) | | 68 (55–79) | |
| Year | | | | | | | | | | | | | | | |
| | 2012 | 7732 | 10.6 | 71 | 16.2 | 37 | 19.7 | 145 | 11.0 | 6318 | 10.8 | 699 | 9.7 | 462 | 8.3 |
| | 2013 | 9835 | 13.5 | 38 | 8.7 | 22 | 11.7 | 162 | 12.3 | 8300 | 14.2 | 792 | 11.0 | 521 | 9.3 |
| | 2014 | 12788 | 17.5 | 66 | 15.1 | 31 | 16.5 | 200 | 15.2 | 10312 | 17.7 | 1197 | 16.6 | 982 | 17.6 |
| | 2015 | 14048 | 19.2 | 66 | 15.1 | 32 | 17.0 | 239 | 18.1 | 11259 | 19.3 | 1288 | 17.8 | 1164 | 20.9 |
| | 2016 | 14204 | 19.4 | 97 | 22.1 | 36 | 19.1 | 268 | 20.3 | 11181 | 19.2 | 1416 | 19.6 | 1206 | 21.6 |
| | 2017 | 14450 | 19.8 | 100 | 22.8 | 30 | 16.0 | 305 | 23.1 | 10937 | 18.8 | 1834 | 25.4 | 1244 | 22.3 |
| Season | | | | | | | | | | | | | | | |
| | Spring | 18330 | 25.1 | 94 | 21.5 | 40 | 21.3 | 329 | 24.9 | 14611 | 25.1 | 1855 | 25.7 | 1401 | 25.1 |
| | Summer | 16058 | 22.0 | 112 | 25.6 | 53 | 28.2 | 303 | 23.0 | 12702 | 21.8 | 1638 | 22.7 | 1250 | 22.4 |
| | Fall | 18082 | 24.8 | 103 | 23.5 | 42 | 22.3 | 340 | 25.8 | 14309 | 24.5 | 1891 | 26.2 | 1397 | 25.0 |
| | Winter | 20587 | 28.2 | 129 | 29.5 | 53 | 28.2 | 347 | 26.3 | 16685 | 28.6 | 1842 | 25.5 | 1531 | 27.4 |
| Weekend | | | | | | | | | | | | | | | |
| | Weekday | 41479 | 56.8 | 263 | 60.0 | 100 | 53.2 | 695 | 52.7 | 32910 | 56.4 | 4160 | 57.6 | 3351 | 60.1 |
| | Weekend | 31578 | 43.2 | 175 | 40.0 | 88 | 46.8 | 624 | 47.3 | 25397 | 43.6 | 3066 | 42.4 | 2228 | 39.9 |
| Daytime | | | | | | | | | | | | | | | |
| | Night | 26225 | 35.9 | 130 | 29.7 | 85 | 45.2 | 365 | 27.7 | 22563 | 38.7 | 1715 | 23.7 | 1367 | 24.5 |
| | Day | 46832 | 64.1 | 308 | 70.3 | 103 | 54.8 | 954 | 72.3 | 35744 | 61.3 | 5511 | 76.3 | 4212 | 75.5 |
| Variables | | All | | Place-provider-matrix | | | | | | | | | | | |
| | | | | Trained responder | | | | Family bystander | | | | Layperson bystander | | | |
| | | | | Public | | Home | | Public | | Home | | Public | | Home | |
| | | N | % | N | % | N | % | N | % | N | % | N | % | N | % |
| Metropolis | | | | | | | | | | | | | | | |
| | Non-metropolis | 40762 | 55.8 | 185 | 42.2 | 88 | 46.8 | 810 | 61.4 | 32347 | 55.5 | 3968 | 54.9 | 3364 | 60.3 |
| | Metropolis | 32295 | 44.2 | 253 | 57.8 | 100 | 53.2 | 509 | 38.6 | 25960 | 44.5 | 3258 | 45.1 | 2215 | 39.7 |
| Witness | | | | | | | | | | | | | | | |
| | No | 40138 | 54.9 | 221 | 50.5 | 122 | 64.9 | 383 | 29.0 | 32794 | 56.2 | 3127 | 43.3 | 3491 | 62.6 |
| | Yes | 32919 | 45.1 | 217 | 49.5 | 66 | 35.1 | 936 | 71.0 | 25513 | 43.8 | 4099 | 56.7 | 2088 | 37.4 |
| Bystander CPR | | | | | | | | | | | | | | | |
| | No | 34230 | 46.9 | 119 | 27.2 | 65 | 34.6 | 611 | 46.3 | 27509 | 47.2 | 2987 | 41.3 | 2939 | 52.7 |
| | Yes | 38827 | 53.1 | 319 | 72.8 | 123 | 65.4 | 708 | 53.7 | 30798 | 52.8 | 4239 | 58.7 | 2640 | 47.3 |
| Bystander DEF | | | | | | | | | | | | | | | |
| | No | 71644 | 98.1 | 375 | 85.6 | 149 | 79.3 | 1285 | 97.4 | 57404 | 98.5 | 6968 | 96.4 | 5463 | 97.9 |
| | Yes | 1413 | 1.9 | 63 | 14.4 | 39 | 20.7 | 34 | 2.6 | 903 | 1.5 | 258 | 3.6 | 116 | 2.1 |
| Primary ECG | | | | | | | | | | | | | | | |

*(Continued)*

**Table 1.** (Continued)

| Variables | | All | | Place-provider-matrix | | | | | | | | | |
| | | | | Trained responder | | | | Family bystander | | | | Layperson bystander | | | |
| | | | | Public | | Home | | Public | | Home | | Public | | Home | |
| | | N | % | N | % | N | % | N | % | N | % | N | % | N | % |
| | VF/VT | 12021 | 16.5 | 160 | 36.5 | 30 | 16.0 | 436 | 33.1 | 7238 | 12.4 | 3162 | 43.8 | 995 | 17.8 |
| | PEA | 8849 | 12.1 | 58 | 13.2 | 25 | 13.3 | 208 | 15.8 | 6971 | 12.0 | 911 | 12.6 | 676 | 12.1 |
| | Asystole | 52187 | 71.4 | 220 | 50.2 | 133 | 70.7 | 675 | 51.2 | 44098 | 75.6 | 3153 | 43.6 | 3908 | 70.0 |
| RTI | | | | | | | | | | | | | | | |
| | 0–3 | 3564 | 4.9 | 40 | 9.1 | 9 | 4.8 | 93 | 7.1 | 2713 | 4.7 | 511 | 7.1 | 198 | 3.5 |
| | 4–7 | 32066 | 43.9 | 210 | 47.9 | 82 | 43.6 | 521 | 39.5 | 25870 | 44.4 | 3144 | 43.5 | 2239 | 40.1 |
| | 8–11 | 15823 | 21.7 | 80 | 18.3 | 40 | 21.3 | 317 | 24.0 | 12558 | 21.5 | 1530 | 21.2 | 1298 | 23.3 |
| | 12–15 | 5385 | 7.4 | 29 | 6.6 | 13 | 6.9 | 115 | 8.7 | 4187 | 7.2 | 544 | 7.5 | 497 | 8.9 |
| | 16- | 16219 | 22.2 | 79 | 18.0 | 44 | 23.4 | 273 | 20.7 | 12979 | 22.3 | 1497 | 20.7 | 1347 | 24.1 |
| | Median (q1-q3) | 7 (5–9) | | 6 (4–8) | | 7 (5–9) | | 7 (5–10) | | 7 (5–9) | | 7 (5–9) | | 7 (5–10) | |
| Airway | | | | | | | | | | | | | | | |
| | ETI | 3709 | 5.1 | 25 | 5.7 | 8 | 4.3 | 65 | 4.9 | 2944 | 5.0 | 355 | 4.9 | 312 | 5.6 |
| | SGA | 20139 | 27.6 | 154 | 35.2 | 45 | 23.9 | 393 | 29.8 | 15538 | 26.6 | 2375 | 32.9 | 1634 | 29.3 |
| | BVM | 40600 | 55.6 | 225 | 51.4 | 114 | 60.6 | 679 | 51.5 | 32866 | 56.4 | 3737 | 51.7 | 2979 | 53.4 |
| | PV | 8609 | 11.8 | 34 | 7.8 | 21 | 11.2 | 182 | 13.8 | 6959 | 11.9 | 759 | 10.5 | 654 | 11.7 |
| EMS Defibrillation | | | | | | | | | | | | | | | |
| | No | 55633 | 76.2 | 249 | 56.8 | 146 | 77.7 | 762 | 57.8 | 46789 | 80.2 | 3556 | 49.2 | 4131 | 74.0 |
| | Yes | 17424 | 23.8 | 189 | 43.2 | 42 | 22.3 | 557 | 42.2 | 11518 | 19.8 | 3670 | 50.8 | 1448 | 26.0 |
| ED level | | | | | | | | | | | | | | | |
| | Level 1 | 10347 | 14.2 | 76 | 17.4 | 32 | 17.0 | 233 | 17.7 | 8044 | 13.8 | 1202 | 16.6 | 760 | 13.6 |
| | Level 2 | 35705 | 48.9 | 252 | 57.5 | 87 | 46.3 | 656 | 49.7 | 28432 | 48.8 | 3674 | 50.8 | 2604 | 46.7 |
| | Level 3 | 23571 | 32.3 | 96 | 21.9 | 60 | 31.9 | 381 | 28.9 | 19110 | 32.8 | 2026 | 28.0 | 1898 | 34.0 |
| | Level 4 | 3434 | 4.7 | 14 | 3.2 | 9 | 4.8 | 49 | 3.7 | 2721 | 4.7 | 324 | 4.5 | 317 | 5.7 |
| Outcomes | | | | | | | | | | | | | | | |
| | Survival to discharge | 5010 | 6.9 | 78 | 17.8 | 15 | 8.0 | 204 | 15.5 | 2857 | 4.9 | 1515 | 21.0 | 341 | 6.1 |
| | Good CPC | 3099 | 4.2 | 51 | 11.6 | 9 | 4.8 | 140 | 10.6 | 1615 | 2.8 | 1091 | 15.1 | 193 | 3.5 |

CPR, cardiopulmonary resuscitation; DEF, defibrillation; ECG, electrocardiogram; VF, ventricular fibrillation; VT, ventricular tachycardia; PEA, pulseless electrical activity; RTI, response time interval; ETI, endotracheal insertion; SGA, supraglottic airway; BVM, bag-valve mask; PV, positive ventilation; CPC, cerebral performance category

among the Public-TR, Public-Family and Public-Layperson groups. This result implies that bystander factors are less important in OHCAs occurring in public places. However, in home settings, there were significantly poorer CPR outcomes in the family and layperson bystander groups than in the TR group. [19] This result suggests that we need to supplement the OHCA Home-Family and Home-Layperson groups with additional CPR resources. To improve the poor outcome of OHCA in the Home-Family and Layperson groups, education for family members is essential. More specific education programs for home bystanders, such as elderly individuals or housewives, are needed. Elderly individuals or housewives cannot easily access CPR training as younger employed individuals can. A dispatch-assisted basic life support program, which is a new education protocol, was proposed for home bystanders. In South Korea, if the EMS dispatcher suspects cardiac arrest via phone call, the dispatcher gives instruction about how to perform CPR until EMS providers arrive at the scene. [20]

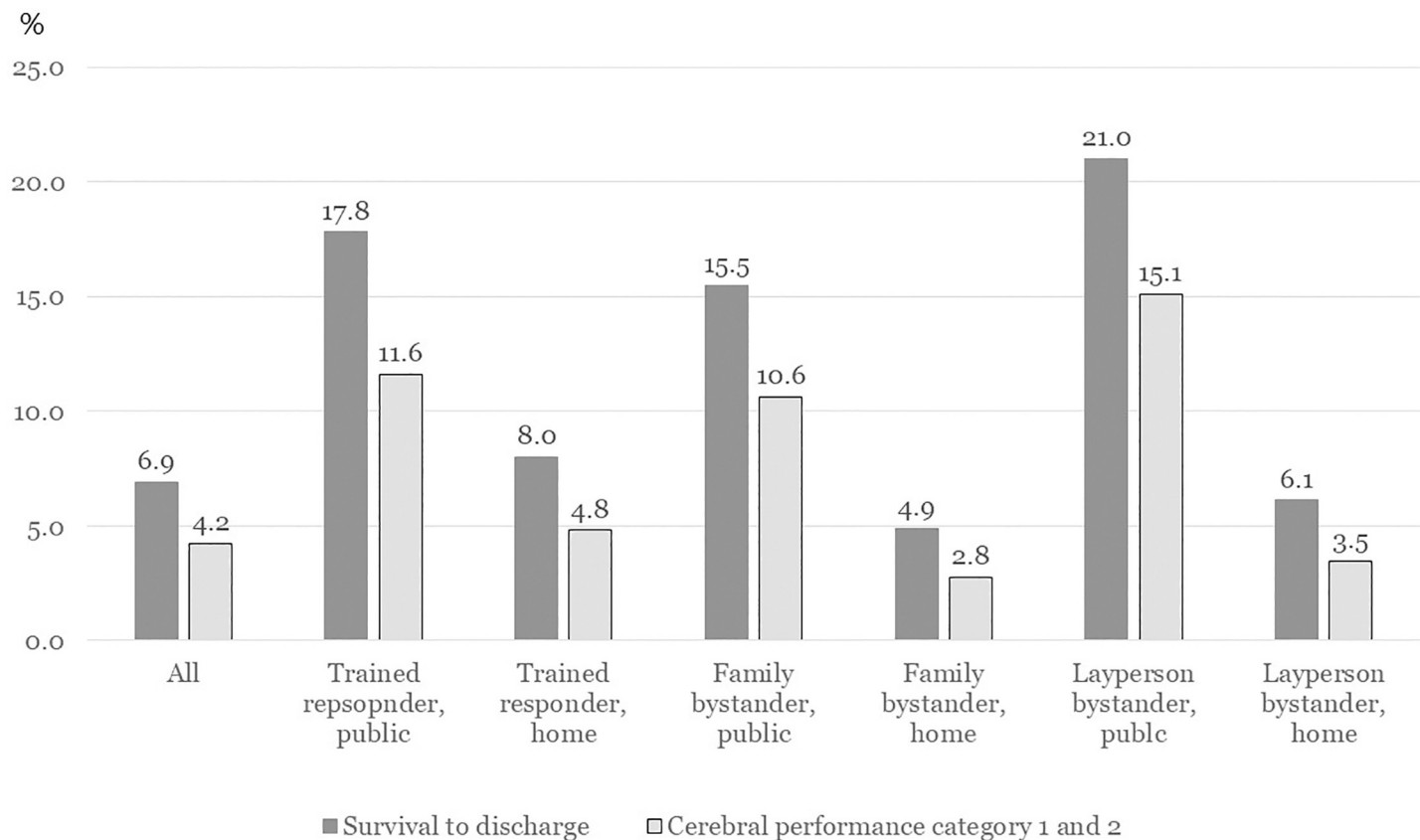

**Fig 2. Outcomes of out-of-hospital cardiac arrest among place-provider matrix group of bystanders.**

Other previous studies reported that family members or friends of patients who were admitted to the hospital due to heart-related disease had better self-confidence after CPR training. Family members who could not participate in hospital CPR education were more motivated to attend hospital CPR training. [21, 22] This concept must be extended to populations with other CPR risk factors, such as hypertension, diabetes mellitus, and hyperlipidemia, so that family members can be encouraged to attend the CPR training program.

In the Public-TR and Home-TR groups, the percentage of both outcomes appeared to be better in the Public-TR group. However, after adjusting for confounders, there was no significant difference between the groups. This means that the Home-TR group and the Public-TR group had equivalent outcomes. This finding proposes that a new CPR program using TR is needed to improve the outcome of OHCA occurring at home. If the place of OHCA is a public site with a high bystander CPR rate and good CPR outcomes in the past, novel dispatcher-assisted CPR instruction may be beneficial. [23] In contrast, if OHCA occurs in private places with potentially low bystander CPR rates and worse outcomes, the EMS dispatcher can activate available TRs, such as town safety guards, policemen, or community security officers near the event place to provide early CPR and defibrillation. The program to activate available TRs or general bystanders has been implemented in a previous study. [24]

TR programs including firefighters and policemen demonstrated an increase in survival to discharge rate as well as a decrease in call-to-scene time and call-to-defibrillation time. [25, 26] In our study, we included not only policemen and firefighters but also public transportation vehicle drivers, school health teachers, sports facility employees, lifeguards, workplace safety

**Table 2. Demographic findings according to arrest time.**

| Variables | | All | | Event Time | | | |
|---|---|---|---|---|---|---|---|
| | | | | Night | | Day | |
| | | N | % | N | % | N | % |
| All | | 73057 | 100.0 | 26225 | 100.0 | 46832 | 100.0 |
| Place-provider-matrix | | | | | | | |
| | Public-Trained | 438 | 0.6 | 130 | 0.5 | 308 | 0.7 |
| | Home- Trained | 188 | 0.3 | 85 | 0.3 | 103 | 0.2 |
| | Public-Family | 1319 | 1.8 | 365 | 1.4 | 954 | 2.0 |
| | Home-Family | 58307 | 79.8 | 22563 | 86.0 | 35744 | 76.3 |
| | Public-Layperson | 7226 | 9.9 | 1715 | 6.5 | 5511 | 11.8 |
| | Home-Layperson | 5579 | 7.6 | 1367 | 5.2 | 4212 | 9.0 |
| Gender | | | | | | | |
| | Female | 25719 | 35.2 | 8755 | 33.4 | 16964 | 36.2 |
| | Male | 47338 | 64.8 | 17470 | 66.6 | 29868 | 63.8 |
| Age | | | | | | | |
| | Age<65 | 26923 | 36.9 | 10902 | 41.6 | 16021 | 34.2 |
| | 65 = <Age<85 | 36569 | 50.1 | 12353 | 47.1 | 24216 | 51.7 |
| | Age> = 85 | 9565 | 13.1 | 2970 | 11.3 | 6595 | 14.1 |
| | Median (q1-q3) | 71 (58–80) | | 69 (56–79) | | 72 (59–81) | |
| Year | | | | | | | |
| | 2012 | 7732 | 10.6 | 2936 | 11.2 | 4796 | 10.2 |
| | 2013 | 9835 | 13.5 | 3614 | 13.8 | 6221 | 13.3 |
| | 2014 | 12788 | 17.5 | 4574 | 17.4 | 8214 | 17.5 |
| | 2015 | 14048 | 19.2 | 5003 | 19.1 | 9045 | 19.3 |
| | 2016 | 14204 | 19.4 | 5033 | 19.2 | 9171 | 19.6 |
| | 2017 | 14450 | 19.8 | 5065 | 19.3 | 9385 | 20.0 |
| Season | | | | | | | |
| | Spring | 18330 | 25.1 | 6649 | 25.4 | 11681 | 24.9 |
| | Summer | 16058 | 22.0 | 6013 | 22.9 | 10045 | 21.4 |
| | Fall | 18082 | 24.8 | 6324 | 24.1 | 11758 | 25.1 |
| | Winter | 20587 | 28.2 | 7239 | 27.6 | 13348 | 28.5 |
| Weekend | | | | | | | |
| | Weekday | 41479 | 56.8 | 14894 | 56.8 | 26585 | 56.8 |
| | Weekend | 31578 | 43.2 | 11331 | 43.2 | 20247 | 43.2 |
| Metropolis | | | | | | | |
| | Non-metropolis | 40762 | 55.8 | 14293 | 54.5 | 26469 | 56.5 |
| | Metropolis | 32295 | 44.2 | 11932 | 45.5 | 20363 | 43.5 |
| Witness | | | | | | | |
| | No | 40138 | 54.9 | 13494 | 51.5 | 26644 | 56.9 |
| | Yes | 32919 | 45.1 | 12731 | 48.5 | 20188 | 43.1 |
| Bystander CPR | | | | | | | |
| | No | 34230 | 46.9 | 12082 | 46.1 | 22148 | 47.3 |
| | Yes | 38827 | 53.1 | 14143 | 53.9 | 24684 | 52.7 |
| Bystander DEF | | | | | | | |
| | No | 71644 | 98.1 | 25783 | 98.3 | 45861 | 97.9 |
| | Yes | 1413 | 1.9 | 442 | 1.7 | 971 | 2.1 |
| Primary ECG | | | | | | | |
| | VF/VT | 12021 | 16.5 | 4632 | 17.7 | 7389 | 15.8 |

*(Continued)*

**Table 2.** (Continued)

| Variables | | All | | Event Time | | | |
|---|---|---|---|---|---|---|---|
| | | | | Night | | Day | |
| | | N | % | N | % | N | % |
| | PEA | 8849 | 12.1 | 3132 | 11.9 | 5717 | 12.2 |
| | Asystole | 52187 | 71.4 | 18461 | 70.4 | 33726 | 72.0 |
| RTI | | | | | | | |
| | 0–3 | 3564 | 4.9 | 1164 | 4.4 | 2400 | 5.1 |
| | 4–7 | 32066 | 43.9 | 12119 | 46.2 | 19947 | 42.6 |
| | 8–11 | 15823 | 21.7 | 5428 | 20.7 | 10395 | 22.2 |
| | 12–15 | 5385 | 7.4 | 1756 | 6.7 | 3629 | 7.7 |
| | 16- | 16219 | 22.2 | 5758 | 22.0 | 10461 | 22.3 |
| | Median (q1-q3) | 7 (5–9) | 7 (5–9) | 7 (5–10) | | | |
| Airway | | | | | | | |
| | ETI | 3709 | 5.1 | 1322 | 5.0 | 2387 | 5.1 |
| | SGA | 20139 | 27.6 | 7249 | 27.6 | 12890 | 27.5 |
| | BVM | 40600 | 55.6 | 14510 | 55.3 | 26090 | 55.7 |
| | PV | 8609 | 11.8 | 3144 | 12.0 | 5465 | 11.7 |
| EMS Defibrillation | | | | | | | |
| | No | 55633 | 76.2 | 19695 | 75.1 | 35938 | 76.7 |
| | Yes | 17424 | 23.8 | 6530 | 24.9 | 10894 | 23.3 |
| ED level | | | | | | | |
| | Level 1 | 10347 | 14.2 | 3855 | 14.7 | 6492 | 13.9 |
| | Level 2 | 35705 | 48.9 | 13091 | 49.9 | 22614 | 48.3 |
| | Level 3 | 23571 | 32.3 | 8140 | 31.0 | 15431 | 32.9 |
| | Level 4 | 3434 | 4.7 | 1139 | 4.3 | 2295 | 4.9 |
| Outcomes | | | | | | | |
| | Survival to discharge | 5010 | 6.9 | 2024 | 7.7 | 2986 | 6.4 |
| | Good CPC | 3099 | 4.2 | 1273 | 4.9 | 1826 | 3.9 |

CPR, cardiopulmonary resuscitation; DEF, defibrillation; ECG, electrocardiogram; VF, ventricular fibrillation; VT, ventricular tachycardia; PEA, pulseless electrical activity; RTI, response time interval; ETI, endotracheal insertion; SGA, supraglottic airway; BVM, bag-valve mask; PV, positive ventilation; CPC, cerebral performance category

employees and travel business employees who are likely to witness cardiac arrest in their work place. A greater number of TRs in our study setting was expected to play an important role in improving early bystander response and defibrillation. Equivalent outcomes were observed in both the Public-TR and Home-TR groups. This finding encourages us to designate more potential providers as TRs and to provide regular CPR education and training.

In a previous study, the TR group showed better outcomes than the layperson bystander group regardless of the place of arrest. [6] As a further detailed study, we found similar results, showing that the Home-Family group and the Home-Layperson group had worse outcomes. However, the Public-Layperson group showed a higher rate of survival to discharge and good neurological outcome than the Public-TR group, although the result was not statistically significant in the main analysis after adjusting for confounders. This implies that OHCA in public places is less affected by bystanders than that in home settings. This is because OHCA in public places can be easily witnessed and has a higher chance of early CPR and defibrillation. Furthermore, patients with OHCA in public places would be younger and have fewer medical illnesses since they are able to walk around public places. [27] Likewise, OHCA in public places

**Table 3. Multivariable logistic regression analysis of outcomes by place-provider-matrix.**

| Outcomes | | Total | Positive | | Model 1 | | | Model 2 | | |
|---|---|---|---|---|---|---|---|---|---|---|
| Survival | | N | N | % | AOR | 95% CI | | AOR | 95% CI | |
| | Total | 73057 | 5010 | 6.9 | | | | | | |
| | Public-Trained | 438 | 78 | 17.8 | 1.00 | | | 1.00 | | |
| | Home-Trained | 188 | 15 | 8.0 | 0.46 | 0.25 | 0.83 | 0.61 | 0.35 | 1.05 |
| | Public-Family | 1319 | 204 | 15.5 | 1.05 | 0.78 | 1.40 | 0.85 | 0.62 | 1.17 |
| | Home-Family | 58307 | 2857 | 4.9 | 0.37 | 0.29 | 0.48 | 0.38 | 0.29 | 0.50 |
| | Public-Layperson | 7226 | 1515 | 21.0 | 1.23 | 0.96 | 1.59 | 1.12 | 0.85 | 1.49 |
| | Home-Layperson | 5579 | 341 | 6.1 | 0.40 | 0.30 | 0.52 | 0.42 | 0.31 | 0.57 |
| Good CPC | | | | | | | | | | |
| | Total | 73057 | 3099 | 4.2 | | | | | | |
| | Public-Trained | 438 | 51 | 11.6 | 1.00 | | | 1.00 | | |
| | Home-Trained | 188 | 9 | 4.8 | 0.45 | 0.22 | 0.94 | 0.58 | 0.27 | 1.25 |
| | Public-Family | 1319 | 140 | 10.6 | 1.18 | 0.83 | 1.66 | 0.88 | 0.61 | 1.27 |
| | Home-Family | 58307 | 1615 | 2.8 | 0.38 | 0.28 | 0.51 | 0.38 | 0.28 | 0.52 |
| | Public-Layperson | 7226 | 1091 | 15.1 | 1.37 | 1.01 | 1.85 | 1.20 | 0.87 | 1.65 |
| | Home-Layperson | 5579 | 193 | 3.5 | 0.38 | 0.27 | 0.53 | 0.42 | 0.30 | 0.59 |

AOR, adjusted odds ratio; 95% CI, 95% confidence interval; CPC, cerebral performance category

Model 1: adjusted for gender and age

Model 2: adjusted for gender, age, year, season, weekend, daytime, metropolis, and witness

minimizes the beneficial effect of bystanders. Further research is needed to prove the detailed association between arrest location and bystander characteristics.

We found a circadian variation in CPR outcomes, which has been reported by many previous studies. [28–30] The exact mechanism of circadian variation in OHCA outcomes has not yet been discovered. However, there is a general consensus that patient activity or environmental factors influence the outcome rather than the circadian variation in the underlying diseases. Our study analyzed the effect of PPM groups according to event time to compare the effect size of circadian factors. From this study, we found that the PPM consistently contributed to outcomes of OHCA regardless of the event time. However, a significantly different effect size in the PPM group on outcomes according to the time of the event was only observed

**Table 4. Multivariable logistic regression analysis for outcomes by arrest time.**

| Outcomes | | Total | Positive | | Model 1 | | | Model 2 | | |
|---|---|---|---|---|---|---|---|---|---|---|
| Survival | | N | N | % | AOR | 95% CI | | AOR | 95% CI | |
| | Total | 73057 | 5010 | 6.9 | | | | | | |
| | Nighttime | 26225 | 2024 | 7.7 | 1.00 | | | 1.00 | | |
| | Daytime | 46832 | 2986 | 6.4 | 0.92 | 0.86 | 0.97 | 0.85 | 0.80 | 0.91 |
| Good CPC | | | | | | | | | | |
| | Total | 73057 | 3099 | 4.2 | | | | | | |
| | Nighttime | 26225 | 1273 | 4.9 | 1.00 | | | 1.00 | | |
| | Daytime | 46832 | 1826 | 3.9 | 0.92 | 0.85 | 0.99 | 0.84 | 0.77 | 0.91 |

CPC, cerebral performance category; AOR, adjusted odds ratio; 95% CI, 95% confidence interval

Model 1: adjusted for gender, age and daytime

Model 2: adjusted for gender, age, daytime, year, season, weekend, place-provider matrix group, metropolis, and witness

**Table 5. Interaction effect between place-provider-matrix and arrest time.**

| Outcomes | | Nighttime | | | Daytime | | |
|---|---|---|---|---|---|---|---|
| Survival | | AOR | 95% CI | | AOR | 95% CI | |
| | Public-Trained | 1.00 | | | 1.00 | | |
| | Home-Trained | 0.79 | 0.47 | 1.32 | 1.19 | 0.68 | 2.08 |
| | Public-Family | 1.28 | 1.06 | 1.53 | 1.10 | 0.90 | 1.35 |
| | Home-Family | 0.60 | 0.53 | 0.68 | 0.48 | 0.42 | 0.55 |
| | Public-Layperson | 1.69 | 1.48 | 1.93 | 1.61 | 1.40 | 1.85 |
| | Home-Layperson | 0.67 | 0.57 | 0.79 | 0.66 | 0.56 | 0.78 |
| Good CPC | | | | | | | |
| | Public-Trained | 1.00 | | | 1.00 | | |
| | Home-Trained | 0.85 | 0.46 | 1.57 | 1.09 | 0.53 | 2.26 |
| | Public-Family | 1.30 | 1.05 | 1.62 | 1.27 | 0.99 | 1.62 |
| | Home-Family | 0.58 | 0.50 | 0.68 | 0.46 | 0.39 | 0.55 |
| | Public-Layperson | 1.74 | 1.48 | 2.03 | 1.73 | 1.45 | 2.07 |
| | Home-Layperson | 0.60 | 0.50 | 0.73 | 0.64 | 0.52 | 0.80 |

CPC, cerebral performance category; AOR, adjusted odds ratio; 95% CI, 95% confidence interval

Interaction model: adjusted for gender, age, daytime, year, season, weekend, metropolis, witness, and interaction term (place-provider-matrix*daytime)

in the Public-Family group with OHCA at night. It is well known that OHCA at night shows worse outcomes than that occurring in the daytime, but previous studies on circadian variations in OHCA have not analyzed the association between arrest time, bystander and place of arrest. Recent research on circadian differences in OHCA reported that there is no relationship between OHCA outcomes and arrest time. [31] This indicates that further study is required to analyze the multifactorial effect of circadian variation in OHCA.

The PPM analysis revealed that private places are associated with a higher risk of poor OHCA outcomes. However, the risk can be reduced by changing the bystander factor from family or layperson to a trained responder. Further study is needed to determine whether an extended TR program in private places and changes in dispatch protocol for activating TRs can reduce the hazardous effect of private places on outcomes of OHCA.

## Limitations

This study has several limitations. First, there is selection bias resulting from the inclusion of only adults with a specific cardiac arrest origin. The inclusion of other populations and individuals with different causes of arrest could have affected the outcomes. Second, exposure variables were collected from the EMS registry. The EMS providers might have received variable information based on the CPR location. This process could have been affected by measurement bias. However, we were unable to test the reliability of the measurements. Although there is a quality assurance program in the fire department for data collection and registry documentation, interrater reliability must be considered. Third, the outcomes (survival to discharge, good CPC) were retrospectively collected from the hospital medical record. This process might have caused detection bias during the medical record review. Furthermore, neurological outcome data were derived from a registry rather than clinical follow-up. The limited follow-up could result in potential bias in regard to outcomes. Fourth, there is a data integrity issue because this study used information from both the prehospital registry and the hospital registry. Fifth, this study was performed in a study setting with a different levels of EMS service. CPR protocols and available medications at the prehospital stage would be different from

those in North American and European countries according to national legislation and the local EMS Act. The lack of comparability of the EMS system to other countries would limit international applicability. This generalization issue must be considered in order to appropriately interpret this study. Last, the difference in the absolute number of patients among PPM groups might have affected the statistical analysis and influenced the results.

## Conclusion

OHCAs that occurred in the home setting with family and layperson bystander groups showed worse outcomes than those of the Public-TR group. The outcomes among place-provider groups were similar regardless of the time of the OHCA event. The findings suggest that OHCAs occurring at home with family or laypersons require a new strategy, such as expanding the TR program to cover more occupations or dispatching nearby TRs to home settings prior to EMS arrival, as an intermediate step to improve CPR outcomes.

## Acknowledgments

This study was supported by the National Emergency Management Agency of Korea and the Korea Centers for Disease Control and Prevention (CDC).

## Author Contributions

**Conceptualization:** Sang Do Shin.

**Funding acquisition:** Sang Do Shin.

**Investigation:** Kyoung Jun Song, Ki Jeong Hong.

**Methodology:** Young Sun Ro, So Yeon Joyce Kong.

**Supervision:** Sang Do Shin.

**Writing – original draft:** Dae Kon Kim.

**Writing – review & editing:** Kyoung Jun Song, Ki Jeong Hong, So Yeon Joyce Kong.

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
