## [Decision Letter · Decision Letter 0]

27 Aug 2019

PONE-D-19-14833

Place-Provider-Matrix of Bystander and Outcomes of Out-of-hospital Cardiac Arrest: A Nationwide Observational Cross-Sectional Analysis

PLOS ONE

Dear Dr. Shin,

Thank you for submitting your manuscript to PLOS ONE. After careful consideration, we feel that it has merit but does not fully meet PLOS ONE’s publication criteria as it currently stands. Therefore, we invite you to submit a revised version of the manuscript that addresses the points raised during the review process.

We would appreciate receiving your revised manuscript by Oct 11 2019 11:59PM. To enhance the reproducibility of your results, we recommend that if applicable you deposit your laboratory protocols in protocols.io, where a protocol can be assigned its own identifier (DOI) such that it can be cited independently in the future. For instructions see: http://journals.plos.org/plosone/s/submission-guidelines#loc-laboratory-protocols

We look forward to receiving your revised manuscript.

Kind regards,

Andrea Ballotta

Academic Editor

PLOS ONE

Journal Requirements:

Additional Editor Comments:

Thank you for your contribution. Your manuscript needs major revision as stated by the two reviewers.

Reviewers' comments:

Reviewer's Responses to Questions

**Comments to the Author**

1. Is the manuscript technically sound, and do the data support the conclusions?

Reviewer #1: No

Reviewer #2: Partly

2. Has the statistical analysis been performed appropriately and rigorously? 

Reviewer #1: No

Reviewer #2: Yes

3. Have the authors made all data underlying the findings in their manuscript fully available?

Reviewer #1: Yes

Reviewer #2: Yes

4. Is the manuscript presented in an intelligible fashion and written in standard English?

Reviewer #1: No

Reviewer #2: Yes

5. Review Comments to the Author

Reviewer #1: please see attachment.

I have to apologize,I forgot to finish this following sentence:

- Line 247: the authors state that the home TR group shows equivalent outcomes with the Public-TR group. In table…

should be continued with:

...In table 3, outcome with good CPC in the Public trained group is 11.3% compared to the outcome in the home-trained which is 4.8%. Please clarify your statement.

Reviewer #2: The data analysis at row 203-206 show a best result in survive/cpc for in public/layperson (20,9%/15,0%) respect in public-TR (17,5%/11,3%), but in the conclusions the authors do not discuss this data, and it is strange that in public place layperson had best results than TR; this data do not support the BLSD Training policy.

Is not easy have a clear vision of all the data, may be useful introduce a graphic representation of the survive/cpc ratio in the different group (bar graphic for example).

In “main analysis” at row 210-217 is not clearly explained (for who is not confident with statistic analysis) what is “AORs” and if is better an high value (0,82) or a low value (0,38), for correct interpretation of the data analysis.

Limitations: another limitation is the high difference of absolute number in the group (for example 187 in Home-TR vs 58264 in Home-Family), that may influence the final conclusions.

Table 1 and table 2 are very “long” and is better rewrite the first row (variable, public, home etc) to the top of every page of the table.

6. PLOS authors have the option to publish the peer review history of their article (what does this mean?). If published, this will include your full peer review and any attached files.

Reviewer #1: No

Reviewer #2: No

---

## [Author Response · Author response to Decision Letter 0]

4 Dec 2019

Author’s reply to reviewers' comments:

On behalf of authors, thank you for the very valuable comments by the reviewers on our paper. We have attempted to address every point commented on by reviewers in the revised manuscript. While we believe that we have addressed all of the reviewers’ concerns, we would be more than pleased to write additional revisions if needed.

We highlighted all changes in red. Author’s answers or explanations are in blue.

Correspondent author

Sang Do Shin, MD, PhD

 

Ms. Ref. No.: PONE-D-19-14833

Title: Place-Provider-Matrix of Bystander and Outcomes of Out-of-hospital Cardiac Arrest : A Nationwide Observational Cross-Sectional Analysis 

Reviewers' comments: 

Reviewer: Professor Andrea Ballotta 

Major concerns:

Overall, the manuscript would profit from a thorough grammatical revision by an English native speaker.

Statistics:

The authors state in the results and the discussion section, that CPR at a public place with trained responders (TR) shows the highest OR of survival. In contrast, in Table 1 and table 3, survival to discharge after CPR of TR in public with good CPC is 11.3%, whereas survival after layperson bystander CPR in public is 15%. - Please clarify, report these results in table 1 by comparing different percentage including statistical differences (p). Also, the whole discussion has to be adapted.

(ANSWER) Thank you for the review and comments. We edited hypothesis (Line 102-103) and discussion (Line 277-287) for showing higher rate of outcomes in the Public-Layperson bystander. Also, we added p-value to the table 1.

(Revision: Introduction-hypothesis)

We hypothesized that the Public-TR group would show the best effect on survival and good neurologic outcome and different PPM groups would show worse outcomes.

(Revision: Discussion)

In the previous study, TR group showed better outcome compared to layperson bystander regardless of arrest place.(6) As a further detailed research, we found similar result showing Home-Family group and Home-Layperson group with worse outcome. However, the Public-Layperson group showed higher rate of survival to discharge and good neurological outcome compared to the Public-TR group, although the result was not statistically significant in main analysis after adjusting confounders. It implies that OHCA in public place is less affected by bystander than home place. This is because OHCA at public place can be easily witnessed and has higher chance of early CPR and defibrillation. Furthermore, patients with OHCA in public place would be younger and have less medical illness to be able to walk around public place.(19) Likewise, OHCA in public place minimizes the beneficial effect of bystander. Further research is needed to prove detailed association between arrest place and bystander characteristic.

Is the term “trained responders/TR” a new definition created by the authors? Please clarify

(ANSWER) Thank you for the review and comments. Our research team has published previous paper about the effect of bystander difference between trained responder and layperson. (Reference 6) This can be a new concept but many other researchers have consensus about necessity of trained first responder. (Reference 26,27)

6. Park YM, Shin SD, Lee YJ, Song KJ, Ro YS, Ahn KO. Cardiopulmonary resuscitation by trained responders versus lay persons and outcomes of out-of-hospital cardiac arrest: A community observational study. Resuscitation. 2017;118:55-62.

26. Hasselqvist-Ax I, Nordberg P, Herlitz J, Svensson L, Jonsson M, Lindqvist J, et al. Dispatch of Firefighters and Police Officers in Out-of-Hospital Cardiac Arrest: A Nationwide Prospective Cohort Trial Using Propensity Score Analysis. J Am Heart Assoc. 2017;6(10).

27. Stein P, Spahn GH, Muller S, Zollinger A, Baulig W, Bruesch M, et al. Impact of city police layperson education and equipment with automatic external defibrillators on patient outcome after out of hospital cardiac arrest. Resuscitation. 2017;118:27-34.

Line 78- It is not clear, if TRs are called to the scene as first responders or are already on the

scene by chance and perform as bystanders. Please clarify.

(ANSWER) Thank you for the review and comments. We added brief explanation of trained responder in Line 83-84. Trained responder can be called to the scene as first responders or already on the scene by chance and perform as bystanders because of characteristics of their job. 

(Revision: Introduction)

Trained responder (TR) refers to the specific population group who has high chance to encounter cardiac arrest or be called to the event scene in daily life in a community.

Per Utstein definition 1, first responders like firefighters or police officers are not "bystanders", since they are part of an organized emergency response system. Therefore, the used definition of “3 types of bystanders” (line 79) should be revised if part of the first responder system.

(ANSWER) Thank you for the review and comments. In South Korea, although trained responders are designated as first responders to encourage bystander CPR by EMS act, TRs do not have legal responsibility for not performing CPR bystander in reality. As we can see in Table 1 and reference 6, CPR cases performed by TR group are extremely low. This study reflected the reality of EMS system in South Korea and that’s why we have divided bystanders into three groups (TR, family, layperson) considering the reality. We added brief explanation about EMS reality in South Korea in Line 86-87)

(Methods: Introduction)

However, TR systems as a component of community CPR program are not well established as part of EMS system in many countries. 

Minor concerns:

Abstract:

Line 55. Please add an explanation for the abbreviation “FR” used in line 68.

Please rephrase the conclusion of the abstract.

(ANSWER) Thank you for the review and comments. We changed sentences as follows. (Line 69-70)

(Revision: Abstract)

The OHCA outcomes of the Home-Family and Home-Layperson group were worse than that of the Public-TR group.

Introduction:

Lines 76-77 are unclear / should be revised by a native speaker.

(ANSWER) Thank you for the review and comments. We changed sentences as follows. (Line 79-80)

(Revision: Introduction)

Rapid cardiopulmonary resuscitation (CPR) and Public Access Defibrillation (PAD) are the most important components of community CPR programs in treating OHCA.

Lines 86-88 and 91-92 should be re-written (unclear).

(ANSWER) Thank you for the review and comments. We changed sentences as follows (Line 88-91, 93-95)

(Revision: Introduction)

Usually, family or layperson bystanders have few CPR experiences, more fear of performing CPR, and more anxiety regarding legal responsibility compared to TRs.(7, 8) The quality of CPR by layperson may be different according to individual’s self-efficacy in performing CPR. Thus, the quality of CPR cannot be assured compared to that of TRs.(4, 9)

OHCA occurred at public place has characteristics of typically younger age, more ventricular fibrillation (VF) in the initial ECG rhythm, and more bystander CPR attempted.

Is the PPM a definition / concept that the authors have newly created or does it already exist per definition? This should be clarified.

(ANSWER) Thank you for the review and comments. The concept of PPM is defined in the main context. (Line 97-98) This is a new concept created by authors.

Methods:

The authors should re-consider depicting the study as "prospective" - was data in fact collected prospectively for the study purpose between 2012 and 2017? In lines 135-136 it is also stated that data derive from a (retrospectively assessed) registry.

(ANSWER) Thank you for the review and comments. We made a slight mistake. We changed the sentence as follows. (Line 111)

(Revision: Methods)

This is a retrospective, nationwide, multicenter and cross-sectional study using the national OHCA registry from 2012 to 2017.

Lines 112-119: This paragraph should be shortened.

(ANSWER) Thank you for the review and comments. We changed the paragraph as follows. (Line 115-120)

(Revision: Methods)

In this study setting, school teachers, sports instructors, public transportation vehicle drivers, safety guards of national parks, and policemen were designated as first responders to encourage bystander CPR by the national emergency medical service (EMS) Act in 2004. In addition, those private places where the TRs work or reside are designated as mandatory sites for PAD programs in 2008 and 2011 by the EMS Act. This group that requires mandatory CPR education is called trained responder (TR). TRs are required to finish regular two-hour course of CPR training at least once in a year since 2005.(6)

Does the study have a positive vote by an Ethics Committee or only the approval of the "review board" of the study hospital?

(ANSWER) Thank you for the review and comments. This study was only approved by the institutional review board of the study hospital. Informed consent was waived. This is mentioned in Line 147-149.

(Revision: Methods)

This study was reviewed and approved by the institutional review board of the study hospital. Informed consent was waived because the data variables did not include personal information, and the study process did not cause any risk for patients. (6) The Korea CDC approved the use of national registry for this study.

What exactly was the pool for patient inclusion - was it access to a nationwide database as described in lines 135-136? This is unclear since expressions such as "the study hospital" are repeatedly used throughout the manuscript (this would suggest only one study centre) and since it seems the study only has "approval of the review board of the study hospital" (would it not be necessary to obtain a nationwide approval by multiple Ethic Committees according to GCP?).

(ANSWER) Thank you for the review and comments. The national OHCA registry has developed and constructed by the Korea Centers for Disease Control and Prevention (CDC) on the basis of National Statistics Act since 2008. All data collection of OHCA were approved by one university institutional review board with a waiver of patients consent because the data was collected from hospital medical record review for index case (OHCA) in the electronic EMS database of the national fire department. After deidentification process, the yearly database is completed to be stored at the Korea CDC electronic data server. If a researcher wants to use the data to do a study, the researcher should obtain an approval from the institutional review board (usually researchers’ affiliated institution) and submit the application for data use to the Korea CDC. The Korea CDC review the application and approve the data use on the basis of review guideline for national registry study. 

Why were only patients over the age of 19 years included in the final analysis? If this is due to local requirements, please specify.

(ANSWER) Thank you for the review and comments. We tried to include only adult patients with cardiac arrest because pediatrics and adolescents have different physiologic status and cardiac arrest features to adult cardiac arrest. There were no local requirements for including only adult patients. 

Line 171. Please add an appropriate reference to “Utstein factors” (see above).

(ANSWER) Thank you for the review and comments. We added an appropriate reference as you have recommended.

(Revision: Methods)

Utstein factors included metropolis, witness, bystander CPR and defibrillation, and primary ECG (shockable versus non-shockable).(16)

Discussion:

Line 232 [... previous research outcomes in other studies]: Please cite and give examples.

(ANSWER) Thank you for the review and comments. We added an appropriate reference as you have recommended. (Line 237-238)

(Revision: Discussion)

Our study findings also correspond to previous research outcomes in other studies.(19)

Given your discussion about poorer outcomes in the group with CA occurring at home, more

literature comparison should be implemented and therefore strengthen your arguments.

(ANSWER) Thank you for the review and comments. We added an appropriate reference as you have recommended. (Line 238-240)

(Revision: Discussion)

The OHCAs occurred in public places showed better survival to discharge and good CPC rate regardless of bystander group. However, in home places, there was significantly poor CPR outcomes in family and layperson bystander groups than TR group.(20)

Line 240 and following paragraphs: You mention dispatcher-assisted BLS - please comment

on the current situation in your country - does it already happen or do you just propose it?

(ANSWER) Thank you for the review and comments. We added brief explanation about dispatcher-assisted BLS status in South Korea. We also added proper reference. Detailed information is written in the reference 21.

(Revision: Discussion)

In South Korea, if EMS dispatcher suspects cardiac arrest via phone call, dispatcher gives CPR instruction about how to do CPR until EMS providers arrive at the scene.(21)

In fact, it is not true that previous studies could not show benefits of e.g. police-first responder systems as stated in line 256. The cited study by Husain et al. explicitly states that in their collective, "survival from out-of-hospital cardiac arrests increased with the implementation of police AED programs".

(ANSWER) Thank you for the review and comments. We edited the paragraph as follows and added appropriate references. (Line 268-276)

(Revision: Discussion)

TR programs including firefighters and policeman demonstrated an increase in survival to discharge rate as well as a decrease in call-to-scene time and call-to-defibrillation time.(26, 27) In our study, we included not only policeman and firefighters but also public transportation vehicle drivers, school health teachers, sports facility employees, lifeguards, workplace safety employees and travel business employees who are likely to witness cardiac arrest in their work place. An extended number of TRs in our study setting was expected to play an important role for improvement of early bystander response and defibrillation. Equivalent outcomes were observed in both Public-TR and Home-TR group. This finding encourages us to designate more potential providers as TR and to provide regular CPR education and training. 

Line 247: the authors state that the home TR group shows equivalent outcomes with the Public-TR group. In table 3, outcome with good CPC in the Public trained group is 11.3% compared to the outcome in the home-trained which is 4.8%. Please clarify your statement.

(ANSWER) Thank you for the review and comments. We changed the sentence as follows. (Line 257-259)

(Revision: Discussion)

In the public-TR and Home-TR group, the percentage of both outcomes seemed like that the public-TR group showed better outcomes. However, after adjusting confounders, it showed no statistical significance between both groups. This means that the home-TR group showed equivalent outcomes with the public-TR group.

Line 250/251: sentence not clear, please rephrase.

(ANSWER) Thank you for the review and comments. We changed the sentence as follows. (Line 261-263)

(Revision: Discussion)

If the place of OHCA is public site where showed high bystander CPR rate and good CPR outcomes in the past, a novel dispatcher-assisted CPR instruction may be beneficial.(24)

Line 265: Please add more up-to-date references about the topic of circadian variation of OHCA.

(ANSWER) Thank you for the review and comments. We added proper references as recommended. (Line 288)

(Revision: Discussion)

We found a circadian variation of CPR outcomes which has been reported by many previous studies.(28-30)

Concerning the day/night findings, the statement in lines 270-271 [A significantly...] is

unclear and needs to be rewritten.

(ANSWER) Thank you for the review and comments. We changed the statement as follows. (Line 293-299)

(Revision: Discussion)

From this study, we found the PPM consistently contributed to outcomes of OHCA regardless of event time of OHCA. However, statistically different effect size of PPM group on outcomes according to time of the event was only observed in the Public-Family group with OHCA at nighttime. It is well-known that OHCA at night shows worse outcome than daytime, but the previous researches about circadian variation of OHCA has not analyzed the association between arrest time, bystander and arrest place as far as we know. This indicates further study is required to analyze multi-factorial effect of circadian variation in OHCA.

Please state the - in your opinion - novelty of your findings.

(ANSWER) Thank you for the review and comments. We stated the importance of this study result as follows. (Line 300-304)

(Revision: Discussion)

The PPM analysis has revealed that private place is exposed to higher risk of poor OHCA outcomes. However, the risk can be reduced by changing bystander factor from family or layperson to trained responder. Further study is needed whether an extended TR program into the private place and change of dispatch protocol for activating TR can reduce the hazardous effect of private place on outcomes and improve outcomes of OHCA or not.

Limitations:

Are patients also followed-up clinically until hospital discharge as part of your registry? Or did you only - as stated - deduct CPC-levels from general documentation such as discharge papers? This is susceptible to considerable bias and should be emphasized more, also in the "Methods" section.

(ANSWER) Thank you for the review and comments. As mentioned before, this study is a registry study. The trained medical record reviewers hired by the Korea CDC evaluate all hospital records including ED care, ICU care and outcome status at patients discharge. This process is described in Data source and collection in Methods part. (Line 141-146) 

Why would "CPR protocols and medications be very different from North America or

Europe" (lines 291-292) when you stated before you adhere to current international CPR guidelines?

(ANSWER) Thank you for the review and comments. Korean EMS providers cannot use epinephrine in CPR situation without medical director’s indication by EMS Act. Although the situation is slowly changing with tremendous effort of EMS physicians, there are many legal and administrative hurdles to catch up the level of EMS providers in North America or European countries. We added a brief explanation about different EMS environment in South Korea.

(Revision: Discussion)

CPR protocols and available medications at prehospital stage would be different from North American and European countries according to national legislation and local EMS act.

Reviewer #2

The data analysis at row 203-206 show a best result in survive/cpc for in public/layperson (20,9%/15,0%) respect in public-TR (17,5%/11,3%), but in the conclusions the authors do not discuss this data, and it is strange that in public place layperson had best results than TR; this data do not support the BLSD Training policy.

(ANSWER) Thank you for the review and comments. We applied your comment and changed the sentence as follows. (Line 239-241)

(Revision: Discussion)

This fact was proven in our study that there was no statistical difference in AORs among public-TR, public-family and public-layperson groups. This result implies that bystander factor is less important in OHCA occurred in public places.

Is not easy have a clear vision of all the data, may be useful introduce a graphic representation of the survive/cpc ratio in the different group (bar graphic for example).

(ANSWER) Thank you for the review and comments. We added Fig 2 for graphical explanation. (Please see Figure 2) The ratios (good CPC/survival) looks similar, which were not presented in the Figure 2. The ratios were as follows; All (0.6), Trained responder, public (0.6), Trained responder, home (0.6), Family bystander, public (0.7), Family bystander, home (0.6), Layperson bystander, public (0.7), and Layperson bystander, home (0.6).

(Revision: Result)

Please see Figure 2.

In “main analysis” at row 210-217 is not clearly explained (for who is not confident with statistical analysis) what is “AORs” and if is better a high value (0,82) or a low value (0,38), for correct interpretation of the data analysis.

(ANSWER) Thank you for the review and comments. We applied your comment and changed the sentence as follows. (Line 216-217, 220-221)

(Revision: Result)

The home-family and home-layperson group had statistically worse result than public-TR group for survival to discharge. 

The home-family and home-layperson group also had statistically worse result than public-TR group for good CPC.

Limitations: another limitation is the high difference of absolute number in the group (for example 187 in Home-TR vs 58264 in Home-Family), that may influence the final conclusions.

(ANSWER) Thank you for the review and comments. We applied your comment and changed the sentence as follows. (Line 321-322)

(Revision: Limitation)

Lastly, the difference of absolute number of patients among PPM groups might have affected statistical analysis and influenced the results.

Table 1 and table 2 are very “long” and is better rewrite the first row (variable, public, home etc) to the top of every page of the table.

(ANSWER) Thank you for the review and comments. The editorial office will revise the tables based on the publishing form after finishing the revision.

---

## [Decision Letter · Decision Letter 1]

15 Jan 2020

PONE-D-19-14833R1

Place-Provider-Matrix of Bystander and Outcomes of Out-of-hospital Cardiac Arrest: A Nationwide Observational Cross-Sectional Analysis

PLOS ONE

Dear Dr. Shin,

Thank you for submitting your manuscript to PLOS ONE. After careful consideration, we feel that it has merit but does not fully meet PLOS ONE’s publication criteria as it currently stands. Therefore, we invite you to submit a revised version of the manuscript that addresses the points raised during the review process.

We would appreciate receiving your revised manuscript by Feb 29 2020 11:59PM. To enhance the reproducibility of your results, we recommend that if applicable you deposit your laboratory protocols in protocols.io, where a protocol can be assigned its own identifier (DOI) such that it can be cited independently in the future. For instructions see: http://journals.plos.org/plosone/s/submission-guidelines#loc-laboratory-protocols

We look forward to receiving your revised manuscript.

Kind regards,

Andrea Ballotta

Academic Editor

PLOS ONE

Additional Editor Comments (if provided):

Dear authors thank you for your efforts. The manuscript looks better although it needs some other improvement. Take in to account the first reviewer's comment especially and go promply to the revision of the paper.

Reviewers' comments:

Reviewer's Responses to Questions

**Comments to the Author**

1. If the authors have adequately addressed your comments raised in a previous round of review and you feel that this manuscript is now acceptable for publication, you may indicate that here to bypass the “Comments to the Author” section, enter your conflict of interest statement in the “Confidential to Editor” section, and submit your "Accept" recommendation.

Reviewer #1: (No Response)

Reviewer #2: All comments have been addressed

2. Is the manuscript technically sound, and do the data support the conclusions?

Reviewer #1: Partly

Reviewer #2: Yes

3. Has the statistical analysis been performed appropriately and rigorously? 

Reviewer #1: Yes

Reviewer #2: Yes

4. Have the authors made all data underlying the findings in their manuscript fully available?

Reviewer #1: Yes

Reviewer #2: Yes

5. Is the manuscript presented in an intelligible fashion and written in standard English?

Reviewer #1: No

Reviewer #2: Yes

6. Review Comments to the Author

Reviewer #1: I thank the authors for their thorough revision of their manuscript and the many points that they could clarify and improve through it. However, in my opinion, there are still a few concerns that must be sufficiently addressed in order to render the manuscript eligible for publication.

Major conerns:

- The manuscript (including the abstract and tables/figures) still is in definitive need of a thorough grammatical revision by an English native speaker. It is not fit to be read by an international audience at the moment.

- The matter of the term “trained responder” is still not resolved. According to Utstein guidelines that should be adhered to, first responders are not bystanders. I understand that in your communities, there are trained persons that work in environments that are likely to be place of a CA. If a CA happens there, then I would call the CPR providers “bystanders”, even if they are specially trained. BUT if they are called or dispatched in any way, they are called “first responders”.

- I think what you should focus on is: 1) Introducing a detailed definition and re-writing the paragraphs where this process is described. Also, describe the calling / dispatching process of TRs in detail. At the moment, it is very confusing. 2) Add a sub-group analyses of a) the cases in which the TRs were real “bystanders” and b) the cases where they were called/dispatched to a CA event. By this, a little structure would be added in this rather confusing topic.

- I still do not fully understand the course your study took when it comes to Ethics Votes. Was the whole registry approved by a proper Ethics Board that is in accordance with international good scientific practice guidelines or was your study in particular approved? If your “review boards” are Ethics Boards, please state so. If not, you might have to present a proper Ethics’ Vote.

- It is a clear limitation that neurological outcomes are derived from registry data rather than clinical follow-up, as you explained after my initial question. Please add this to your Limitations section.

- I was not aware of the EMS situation in South Korea, thank you for the insight. However, also this must be added to the Limitations section because due to the limited comparability to other countries (e.g. in Europe or the USA), also your results are of limited applicability internationally.

Minor concerns:

- My request concerning the inclusion of only patients over 19 years aimed at the – in my opinion – international definition of “adults” for people over 18 years and not 19 – please state why you chose 19.

- The sentence “The OHCAs…” (lines 239-240) is still not clear, please re-phrase.

- You might want to add the contradicting results of up-to-date literature to your statement that time-of-day influences outcomes (for example Schriefl C et al., Time of out-of-hospital cardiac arrest is not associated with outcome in a metropolitan area: A multicenter cohort study.Resuscitation 2019).

Reviewer #2: no more comments

7. PLOS authors have the option to publish the peer review history of their article (what does this mean?). If published, this will include your full peer review and any attached files.

Reviewer #1: No

Reviewer #2: No

---

## [Author Response · Author response to Decision Letter 1]

31 Mar 2020

Author’s reply to reviewers' comments:

On behalf of authors, thank you for the very valuable comments by the reviewers on our paper. I am very sorry for late revision. We have been fighting with the COVID-19 pandemic for the last two months.

. 

We have attempted to address every point commented on by reviewers in the revised manuscript. While we believe that we have addressed all of the reviewers’ concerns, we would be more than pleased to write additional revisions if needed. 

We highlighted all changes in red. Author’s answers or explanations are in blue.

Correspondent author

Sang Do Shin, MD, PhD

Ms. Ref. No.: PONE-D-19-14833

Title: Place-Provider-Matrix of Bystander and Outcomes of Out-of-hospital Cardiac Arrest: A Nationwide Observational Cross-Sectional Analysis 

Reviewers' comments: 

Major concerns:

- The manuscript (including the abstract and tables/figures) still is in definitive need of a thorough grammatical revision by an English native speaker. It is not fit to be read by an international audience at the moment.

(ANSWER) Thank you for the review and comments. The manuscript was revised again for grammatical revision by native speakers.

- The matter of the term “trained responder” is still not resolved. According to Utstein guidelines that should be adhered to, first responders are not bystanders. I understand that in your communities, there are trained persons that work in environments that are likely to be place of a CA. If a CA happens there, then I would call the CPR providers “bystanders”, even if they are specially trained. BUT if they are called or dispatched in any way, they are called “first responders”.

(ANSWER) Thank you for the review and comments. We tried to explain the term “trained responder” more specifically in the introduction and method paragraph. We added brief explanation of trained responder in Line 84-88, 118-120 and 124-128. 

(Revision: Introduction, line 84-88)

Trained responders (TRs) refer to a specific group of individuals who have a high chance of encountering cardiac arrest in daily life because of occupational characteristics and training for CPR situations but who are not part of the officially organized emergency response system in a community. In general, TRs fall between a bystander and first responder according to the Utstein definition of provider.

(Revision: Method, line 118-120)

According to the 2004 National Emergency Medical Services (EMS) Act, school teachers, sports instructors, public transportation vehicle drivers, safety guards of national parks, and policemen are required to receive CPR education to encourage bystander CPR.

(Revision: Method, line 124-128)

TRs are not first responders who respond to medical emergencies in an official capacity as part of an organized medical response team. Rather, they are similar to bystanders but have been trained in CPR because of the higher chance of encountering CPR situations due to occupational characteristics. Therefore, TRs do not have the duty of official calls or dispatches from EMS systems. They voluntarily participate in the CPR situation.

- I think what you should focus on is: 1) Introducing a detailed definition and re-writing the paragraphs where this process is described. Also, describe the calling / dispatching process of TRs in detail. At the moment, it is very confusing. 2) Add a sub-group analyses of a) the cases in which the TRs were real “bystanders” and b) the cases where they were called/dispatched to a CA event. By this, a little structure would be added in this rather confusing topic.

(ANSWER) Thank you for the review and comments. The revised paragraphs in the above comment can be a reply for this comment. No different in calling and dispatching process of TRs. Just we can identify the TR groups from the EMS CPR registry. The dispatch for CPR is under the same protocol. You asked a subgroup analysis for the case in which the TRs were “real bystander”. But we have analyzed the bystanders as subgroup such as “Trained responder, Family bystander, and Layperson bystander” The Layperson bystander and Family bystanders are part of general (real) bystanders, not trained bystanders

- I still do not fully understand the course your study took when it comes to Ethics Votes. Was the whole registry approved by a proper Ethics Board that is in accordance with international good scientific practice guidelines or was your study in particular approved? If your “review boards” are Ethics Boards, please state so. If not, you might have to present a proper Ethics’ Vote.

(ANSWER) Thank you for the review and comments. The OHCA database had been constructed by the government par (Korea Centers for Disease Control and Prevention). When a researcher wants to use the database for a research, he or she should get permission of data use using a data request form and Korea CDC review and decide to permit or not the use of data. When a research request the data, he or she also get approval of the IRB at each institution. Our institutional review board has reviewed and approved our specific research.

- It is a clear limitation that neurological outcomes are derived from registry data rather than clinical follow-up, as you explained after my initial question. Please add this to your Limitations section.

(ANSWER) Thank you for the review and comments. We added additional explanation of limitation in Line 327-328.

(Revision: Limitation)

Furthermore, neurological outcome data were derived from a registry rather than clinical follow-up. The limited follow-up could result in potential bias in regard to outcomes.

- I was not aware of the EMS situation in South Korea, thank you for the insight. However, also this must be added to the Limitations section because due to the limited comparability to other countries (e.g. in Europe or the USA), also your results are of limited applicability internationally.

(ANSWER) Thank you for the review and comments. We added additional explanation of limitation in Line 329-333.

(Revision: Limitation)

Fifth, this study was performed in a study setting with a different levels of EMS service. CPR protocols and available medications at the prehospital stage would be different from those in North American and European countries according to national legislation and the local EMS Act. The lack of comparability of the EMS system to other countries would limit international applicability.

Minor concerns:

- My request concerning the inclusion of only patients over 19 years aimed at the – in my opinion – international definition of “adults” for people over 18 years and not 19 – please state why you chose 19.

(ANSWER) Thank you for the review and comments. We changed inclusion criteria age to over 18 years old. The relevant paragraphs of methods, results, and discussion were revised and Tables and Figure were changed due to the criteria.

- The sentence “The OHCAs…” (lines 239-240) is still not clear, please re-phrase.

(ANSWER) Thank you for the review and comments. We re-phrased the sentence in line 248-249.

(Revision: Discussion)

In this study, both outcomes showed higher rates in public places than in home settings, which was independent of bystander type.

- You might want to add the contradicting results of up-to-date literature to your statement that time-of-day influences outcomes (for example Schriefl C et al., Time of out-of-hospital cardiac arrest is not associated with outcome in a metropolitan area: A multicenter cohort study. Resuscitation 2019).

(ANSWER) Thank you for the review and comments. We added additional explanation in line 308-309.

(Revision: Discussion)

Recent research on circadian differences in OHCA reported that there is no relationship between OHCA outcomes and arrest time.

---

## [Editor Report · Decision Letter 2]

28 Apr 2020

Place-Provider-Matrix of Bystander Cardiopulmonary Resuscitation and Outcomes of Out-of-hospital Cardiac Arrest: A Nationwide Observational Cross-Sectional Analysis

PONE-D-19-14833R2

Dear Dr. Shin,

We are pleased to inform you that your manuscript has been judged scientifically suitable for publication and will be formally accepted for publication once it complies with all outstanding technical requirements.

With kind regards,

Andrea Ballotta

Academic Editor

PLOS ONE

Additional Editor Comments (optional):

Congratulations!!!!! "nulla obsta" to the acceptance for publication of your revised manuscript.
---

## [Editor Report · Acceptance letter]

5 May 2020

PONE-D-19-14833R2 

Place-Provider-Matrix of Bystander Cardiopulmonary Resuscitation and Outcomes of Out-of-hospital Cardiac Arrest: A Nationwide Observational Cross-Sectional Analysis 

Dear Dr. Shin:

I am pleased to inform you that your manuscript has been deemed suitable for publication in PLOS ONE. Congratulations! Your manuscript is now with our production department. 

With kind regards,

on behalf of

Dr. Andrea Ballotta 

Academic Editor

PLOS ONE